# SemanticBoost: Elevating Motion Generation with Augmented Textual Cues

## Abstract

Current techniques face difficulties in generating motions from intricate semantic descriptions, primarily due to insufficient semantic annotations in datasets and weak contextual understanding. To address these issues, we present SemanticBoost, a novel framework that tackles both challenges simultaneously. Our framework comprises a Semantic Enhancement module and a Context-Attuned Motion Denoiser (CAMD). The Semantic Enhancement module extracts supplementary semantics from motion data, enriching the dataset's textual description and ensuring precise alignment between text and motion data without depending on large language models. On the other hand, the CAMD approach provides an all-encompassing solution for generating high-quality, semantically consistent motion sequences by effectively capturing context information and aligning the generated motion with the given textual descriptions. Distinct from existing methods, our approach can synthesize accurate orientational movements, combined motions based on specific body part descriptions, and motions generated from complex, extended sentences. Our experimental results demonstrate that SemanticBoost, as a diffusion-based method, outperforms auto-regressive-based techniques, achieving cutting-edge performance on the Humanml3D dataset while maintaining realistic and smooth motion generation quality.

## 1 Introduction

Over recent years, motion generation from textual descriptions has made significant progress Zhang et al. (2023a); Chen et al. (2022); Jiang et al. (2023); Zhang et al. (2023b), enhancing creativity and realism in applications like animation, robotics, and virtual reality. However, generating motion from complex semantic descriptions remains challenging due to the lack of comprehensive semantic annotations in datasets like Humanml3D Guo et al. (2022a) and the limited contextual understanding of existing techniques.

Text annotations in these datasets are often simplistic and less informative. For example, an annotation might state, "A person walks in a straight line," but the actual motion data contains more details, such as head orientation and the direction of the movement, which are not captured in the text. This information gap complicates the generation of accurate and realistic motion from textual descriptions. Additionally, current techniques' limited contextual understanding worsens this problem, as they often disregard global motion cues and word embeddings during the learning process. Instead, they only rely on full-sentence text embeddings, neglecting the importance of understanding individual words' context and their relationships within the sentence, which is crucial for generating more accurate and contextually relevant motion.

To address these challenges, we introduce SemanticBoost, a novel framework designed to tackle both insufficient semantic annotations and weak contextual comprehension in motion generation. SemanticBoost comprises two key components: the Semantic Enhancement module and the Context-Attuned Motion Denoiser (CAMD), which work together to perform motion synthesis from textual descriptions.

The Semantic Enhancement module plays a crucial role by extracting additional details from motion data, enriching the textual descriptions in the dataset. This enhancement ensures accurate alignment between text and motion data without relying on large, resource-intensive language models. Meanwhile, the CAMD approach synthesizes high-quality, semantically consistent motion sequences.

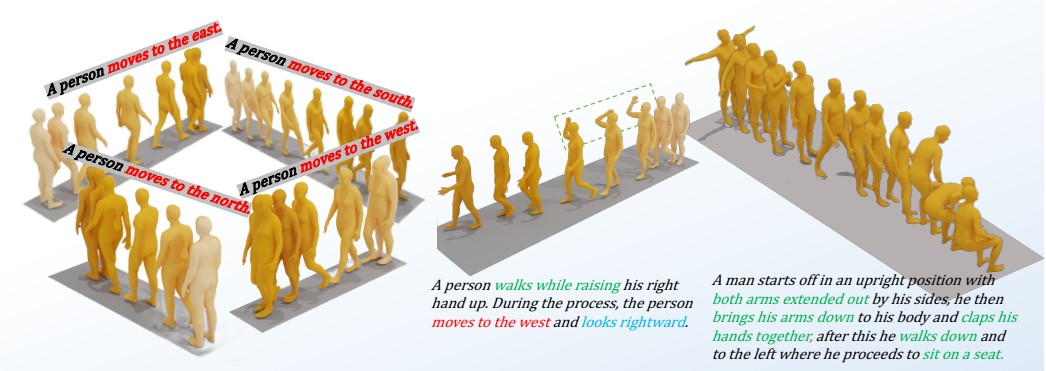

*A person walks while raising his right hand up. During the process, the person moves to the west and looks rightward.*

*A man starts off in an upright position with both arms extended out by his sides, he then brings his arms down to his body and claps his hands together; after this he walks down and to the left where he proceeds to sit on a seat.*

Figure 1: Illustration of Our method's unique capabilities in text-to-motion generation, showcasing precise orientational movements, combined motions based on specific body part descriptions, and motion generation from complex, lengthy sentences. We mark the descriptions of orientational movements with red, the head orientation with blue, and the sub-actions with green.

Unlike previous methods that directly denoise individual noised frames, our method first encodes the motion to denoise via a tiny convolutional network attaching temporal max-pooling to a compact feature vector. It is then concatenated with each noised frame before denoising. This feature encodes the global statistics of the motion to denoise, thus allowing more effective information exchange among frames in the denoising module. We also utilize localized word embeddings from a sentence to establish a stronger connection between motion and semantics through a cross-attention structure. Consequently, our method captures context information and expertly aligns generated motion with the provided textual descriptions.

We rigorously evaluate SemanticBoost's effectiveness through extensive experiments, demonstrating its superiority over existing approaches. Notably, our method is the first diffusion-based approach to outperform autoregressive model-based approaches, achieving state-of-the-art performance on the challenging Humanml3D dataset. Furthermore, as shown in Fig.1, our method excels at synthesizing accurate orientational movements, combined motions based on specific body part descriptions, and motions generated from complex, extended sentences, setting it apart from its counterparts.

## 2 RELATED WORK

### 2.1 HUMAN MOTION GENERATION

Accurate motion signals are widely utilized in many fields, such as film, gaming, live streaming, etc. As a result, there is a long research history on the human motion synthesizing task. There are methods for generating motion "in-between," which are provided with a part of motion clips and fill others Duan et al. (2021); Harvey & Pal (2018); Harvey et al. (2020); Kaufmann et al. (2020); Tang et al. (2022), methods driven by music Lee et al. (2019); Li et al. (2020; 2021; 2022); Aristidou et al. (2021); Chen et al. (2021). This paper will focus on motion generation driven by text Ahuja & Morency (2019); Guo et al. (2022b); Kim et al. (2022); Petrovich et al. (2022), etc

Due to the lack of annotated datasets, there are works that address the problem text to action, such as ACTOR Petrovich et al. (2021) based on large-scale action datasets NTU-RGB Shahroudy et al. (2016) and NTU-RGBD-120 Liu et al. (2020). However, the limited information provided by a single action label leads to inaccurate synthesizing results. Recently, there have been two 3D motion datasets with annotated textual descriptions KIT-ML Plappert et al. (2016) and HumanML3D Guo et al. (2022a). Based on the datasets, there are two kinds of strategies to synthesize 3D motion sequences. The first one is the two-stage autoregressive model-based approach, which learns a latent space at the first stage and then predicts the latent features at the second stage, such as Guo et al. (2022a) and T2M-GPT Zhang et al. (2023a). This strategy achieves great evaluation results, but there are a few jitters in their synthesized motion sequences due to the discrete latent space. Another

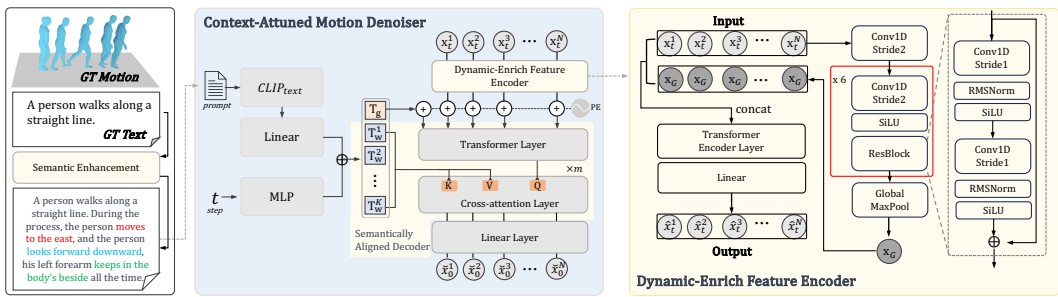

Figure 2: The overview of our SemanticBoost consists of a Semantic Enhancement module, which is illustrated in the left, and a Context-Attuned Motion Denoiser which is illustrated in the middle. The denoiser includes two sub-modules: a Dynamic-Enrich Feature Encoder whose structure is illustrated in the right and a Semantically Aligned Decoder which utilizes the word embeddings to enhance the relationship between synthesized motion sequences and textual descriptions.

strategy is the diffusion-based generative method, such as MDM Tevet et al. (2022), MotionDiffuse Zhang et al. (2022) and faster MLD Chen et al. (2022), which adds noise to motion data and learns a denoiser to synthesize motion from random noise. These diffusion-based approaches demonstrate improved motion continuity in the synthesized results. As an optimization, our method sets itself apart by explicitly extracting holistic motion features and employing word embeddings with cross-attention. This leads to a more accurate alignment between the synthesized motion and the textual descriptions, ultimately capturing context information more effectively

## 2.2 SEMANTIC ANNOTATION AUGMENTATION

Even though there are already textual annotations in the 3D motion datasets KIT-ML and Hu-manML3D, existing annotations are still far from accurately describing the fine details of the actions which will make it difficult to decouple fine-grained control from specific actions. To address the problem, there are several works have tried to enhance semantic annotations utilizing the prior information of large-scale language models such as SINC Athanasiou et al. (2023) which predicts the common details during a motion process according to GPT-3 Brown et al. (2020), action-GPT Kalakonda et al. (2023) which crafts carefully prompts of LLMs and generate richer descriptions for action labels. These methods primarily utilize large language models, either explicitly or implicitly, to enhance text annotations. However, the improved text annotations may not effectively align with the ground-truth motion data. Differently, our approach extracts additional semantic annotations directly from the motion data, which results in a better alignment between generated detailed descriptions and the ground-truth motion and potentially improves the overall performance.

## 3 METHODOLOGY

In this section, we provide a comprehensive description of our proposed SemanticBoost framework, specifically tailored to address the challenges of limited semantic annotations and weak contextual understanding in motion generation. As illustrated in Fig.2, our framework consists of two main components: the Semantic Enhancement module and the Context-Attuned Motion Denoiser (CAMD).

The Semantic Enhancement module is responsible for extracting additional details from motion data, thereby enriching the original textual descriptions available in the dataset. This process ensures more accurate alignment between the text and motion data. On the other hand, the CAMD comprises a Dynamic-Enrich Feature Encoder and a Semantically Aligned Decoder.

The Dynamic-Enrich Feature Encoder module extracts features that represent the entire motion explicitly. These features are then combined with each noised frame before undergoing the denoising process. The Semantically Aligned Decoder utilizes word embedding techniques and reinforces the relationship between motion and semantics through the implementation of cross-attention mechanisms.

### 3.1 SEMANTIC ENHANCEMENT

In this section, we present our Semantic Enhancement module, which consists of motion sequence enhancement and textual description enhancement.

**Motion Sequence Enhancement**  Existing text annotations often lack body direction descriptions, leading to ambiguities during training. Previous works standardize the initial body directions of all motion sequences to the +Z axis, which results in uncontrollable body directions. In contrast, our approach aims to explore more complex control of body directions within the network. To achieve this, we augment each motion sequence by rotating it 0, 90, 180, or 270 degrees around the vertical axis. This method not only increases data diversity but also addresses the ambiguity issue through the following textual description enhancement.

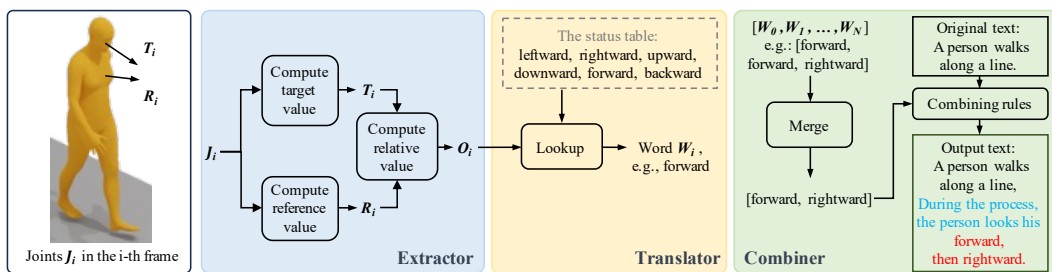

Figure 3: An overview of the textual enhancement pipeline. The pipeline comprises an extractor to obtain essential values from motion data, a translator to convert these values to words, and a combiner to integrate the words with the original text, generating the output text.

**Textual Description Enhancement**  We propose a pipeline to enhance text descriptions with motion details derived directly from the motion data, without relying on external information. These details include body directions, head orientations, hand statuses, and the statuses of any body parts we want to control. As depicted in Fig.3, the pipeline consists of three components: an extractor to obtain crucial values from motion data, a translator to convert these values into words, and a combiner to merge the words with the original text, producing the output text.

(1) Extractor. Taking descriptions of head orientation as an example, given joint positions $J_i$ in the $i$-th frame, the computation in the extractor follows:

$$
\begin{aligned}
T_i &= \frac{1}{2}\left(J_i^{nose} + p^{mid-eye}\right) - p^{mid-ear}, \\
R_i &= \left(J_i^{neck} - J_i^{l-shoulder}\right) \times \left(J_i^{neck} - J_i^{r-shoulder}\right), \\
O_i &= Rodrigues\left(R_i, Z\right) \cdot T_i
\end{aligned}
\tag{1}
$$

where $p^{mid-ear} = \frac{\left(J_i^{l-ear} + J_i^{r-ear}\right)}{2}$ denotes the mid-point of both ears, $p^{mid-eye}$ is defined similarly. $Rodrigues$ represents the Rodrigues' rotation formula, which computes the rotation matrix that rotates $R_i$ to the $Z$ axis. Consequently, $O_i$, a 3-dimensional vector, represents the relative direction of the head concerning the body.

(2) Translator. For each body part, we define a list of statuses as well as the lookup rules to obtain a word $W_i$ given $O_i$. Note that the rules are based on common sense and do not involve any ad-hoc design. For instance, the classification is "rightward" when $O_i^x > 0$, "upward" when $O_i^y > 0$, "forward" when $O_i^z > \mu$, and vice versa, where $\mu > 0$ since the head cannot actually turn backward, and $O_i^z$ is always positive.

(3) Combiner. Given a list of words $[W_0, W_1, ..., W_N]$, we merge all consecutive duplicates before applying the combining rules. These rules use simple templates to integrate the original text with the status words into a complete sentence.

In practice, we down-sample the frames by a factor of 10. In addition to the head, we consider the status of body directions as *east, north, west, south*, the status of both hands as eight horizontal directions relative to the hips, and an additional state "raise up."

### 3.2 CONTEXT-ATTUNED MOTION DENOISER

The enhancement of textual descriptions results in much longer textual descriptions. To keep alignment between motion sequences and descriptions during synthesizing with longer descriptions, we propose a new diffusion denoising model CAMD consists of three parts: Dynamic-Enrich Feature Encoder, Semantically Aligned Decoder, and an additional linear layer. The overview of the model is shown in Fig.2 middle. We represent the input noise frame tokens as $x_t^n$ where $n \in \{1, 2, ..., N\}$ and there are $N$ frames for a motion sequence during the training stage. We represent the sentence embedding and word embeddings by CLIP model as $T_g$ and $T_w^k$. $k \in \{1, 2, ..., K\}$ and $K$ is default length of a sentence. Through our CAMD, denoised or synthesized motion sequences are represented as $\breve{x}_0$.

**Dynamic-Enrich Feature Encoder**   We hypothesize that the suboptimal performance of prior diffusion-based approaches in synthesizing extended motion sequences can be attributed to the inadequate global context provided for tokens embedded within noisy frames. According to previous research, the transformer architecture may require a large amount of data to capture more global information effectively Dosovitskiy et al. (2020).

To address this issue, we propose a Dynamic-Enrich Feature Encoder that embeds the input tokens $x_t$ into a global token $x_G$. Specifically, Given a sequence of frame noise $x_t$, our first step is to pass it through a network structure that utilizes one-dimensional convolution and max pooling. This approach is specifically chosen to capture the global information of the sequence. The reason behind this choice is that a convolutional network based on one-dimensional convolution and max pooling possesses key characteristics, such as hierarchical feature extraction, receptive field expansion, and local temporal relationship capture. These attributes make it particularly well-suited for extracting global information from frame sequences. Then, we combine the extracted global features with each of the original noised frame features. This combined data is then processed through a transformer, which allows the resulting features to integrate both local and global information while also strengthening their temporal relationship. This integration is crucial as it enables the network to comprehend better the overall contextual information related to the action being performed.

**Semantically Aligned Decoder**   Generally, we embed the textual descriptions to tensors by the text encoder of the pre-trained CLIP model. The model converts an entire sentence into a vector and exhibits good generalization capabilities. However, during synthesizing with longer descriptions, we observe that the synthesized results sometimes exhibit inconsistencies in terms of their execution order compared to the provided descriptions or even miss some actions. For this problem, we hypothesize the vector embedded by an entire sentence may lose a part of the original information. As a result, in addition to the whole sentence, we also capture the embeddings of each word $T_w^k$ in the sentence during the inference process of the CLIP model. A layer of Semantically Aligned Decoder consists of two sub-layers. At first, we follow the basic structure of MDM to synthesize the motion sequences with the global token $T_g$ and new frame tokens output by Dynamic-Enrich Feature Encoder. Then, we further align the results with the word embeddings by a cross-attention layer to avoid losing information. Through the structure, we further enhance the alignment between synthesized motion sequences and textual descriptions.

### 3.3 DIFFUSION TRAINING

We follow the basic settings of MDM and synthesize ground-truth motion sequences $x_0$ directly during the training stage. A simple diffusion loss Tevet et al. (2022) for optimizing:

$$\mathcal{L}(x_0, t, p) = \boldsymbol{E}_{x_0 \sim q(x_0|p), t \sim [1,T]}[\|x_0 - G(x_t, t, p)\|_2^2] \tag{2}$$

where $t$ is randomly sampled diffusion step, $p$ means the input prompt, and $G$ is the inference process of the model.

## 4 EXPERIMENTS

In this section, we report our experimental results. In section 4.1, we introduce the basic experimental settings. In section 4.2, we compare our experimental results with state-of-the-art approaches, as well as ablation studies and semantic enhancement evaluation. In section 4.3, we demonstrate the advantages of our method by visualization demos.

### 4.1 BASIC SETTINGS

**Datasets**   Following recent works, we carry out our experiments on two standard text-to-motion datasets.

HumanML3D Guo et al. (2022a) is a 3D human motion-language dataset that originates from a combination of HumanAct12 Guo et al. (2020) and Amass dataset Mahmood et al. (2019). The dataset consists of 14,616 motion sequences and 44,970 descriptions composed of 5,371 words. There are 3-4 descriptions for each motion sequence. The total length of the HumanML3D dataset is about 28.59 hours, and all motion sequences are down-sampled into 20 fps.

KIT Motion-Language(KIT-ML) Plappert et al. (2016) consists of 3,911 human motion sequences and 6,278 textual annotations composed of 1,623 unique words. The motion sequences of the KIT-ML dataset are selected from the KIT dataset Mandery et al. (2016) and the CMU dataset Lab, and all the motion sequences are down-sampled into 12.5 fps.

**Motion Representations**   HumanML3D represents each frame of a motion sequence as a vector with 263 elements and 251 elements for KIT-ML. The vector is represented as $(r^a, r^x, r^z, r^y, j^p, j^r, j^v, c^f)$. $r^a \in \mathbb{R}$ means global root angular velocity, $r^x \in \mathbb{R}$ and $r^z \in \mathbb{R}$ represent global root velocity relative to X and Z axes, respectively. $r^y \in \mathbb{R}$ is the height of root joint. $j^p \in \mathbb{R}^{3(j-1)}$, $j^r \in \mathbb{R}^{6(j-1)}$ and $j^v \in \mathbb{R}^{3j}$ represent position, 6D rotation and velocity of the joints where $j$ is the number of joints. $j = 22$ for HumanML3D and $j = 21$ for KIT-ML. Finally, $c^f \in \mathbb{R}^4$ is a binary vector representing the foot contact information. Though the representation has been valid in recent methods, there are shortcomings during visualization and synthesizing joints with few descriptions in the dataset. To overcome this problem, we replace $j^r \in \mathbb{R}^{6(j-1)}$ with $j^r \in \mathbb{R}^{6j}$ in HumanML3D to get a 269-dimensional vector representation. Specifically, we replace the derived rotation information by joint position with ground-truth rotation information of source data.

**Evaluation Metrics**   Following recent works, we embed motion sequences and textual descriptions into a common subspace with a pre-trained evaluation model. Then, we measure the model quality with five metrics: Frechet Inception Distance (FID) measures the distance between ground truth and synthesized motion sequences. Then, we synthesize 32 motion sequences with one ground-truth description and 31 randomly selected ones. R-precision reports the Top-1, Top-2, and Top-3 accuracy of motion-to-text retrieval. Multi-modal Distance (MM-Dist) measures the average distances between the textual descriptions and their synthesized motion sequences. For Diversity, we randomly sample 300 pairs of motion sequences and compute the average distance of these pairs. Finally, for multi-modality (MModality), we synthesize 30 motion sequences with the same description and randomly choose 10 pairs to compute the average distance of these pairs.

**Additional Evaluation Metrics**   In this paper, we explicitly enrich annotations with translation, head orientation, and left forearm position descriptions. To evaluate the model's control ability over these enhanced parts, we introduced additional metrics except for existing metrics called translation similarity(TS), head orientation similarity (HOS), and left forearm similarity(LFS). We list all possible statuses. Then, we check each frame, divide it into one of these statuses, and finally, we get a statistical vector that represents the distribution of our pointed body part. The additional metrics compare the cosine similarity of the statistical vector between real and synthesized motion.

**Implement Details**   The following implement settings are identical for both datasets. We adopt the AdamW optimizer with parameters [0.9, 0.999], 1000 iterations learning rate warm-up, and cosine schedule cycled per 50K iterations. The nosing steps $T = 1000$ for the diffusion framework, and the nosing scheduler is a cosine scheduler. For the model structure, the default length of a motion sequence $N = 196$, the default length of a textual annotation $K = 77$, the number of transformer

Table 1: Comparison with the state-of-the-art methods on test set of HumanML3D and KIT-ML. We evaluate common evaluation metrics of our whole method on HumanML3D and only report the most representative FID of the basic CAMD model on KIT-ML. † annotates the method that has been re-measured with the evaluation process of T2M-GPT based on provided weights due to abnormal evaluation results of the original paper. We bold the best results for each metric.

| Methods | HumanML3D | | | | | | | KIT-ML |
|---|---|---|---|---|---|---|---|---|
| | R-Precision ↑ | | | FID ↓ | MM-Dist ↓ | Diversity → | MModality ↑ | FID ↓ |
| | Top-1 | Top-2 | Top-3 | | | | | |
| Real | 0.511 | 0.703 | 0.797 | 0.002 | 2.974 | 9.503 | - | 0.031 |
| TM2T Guo et al. (2022c) | 0.424 | 0.618 | 0.729 | 1.501 | 3.467 | 8.589 | 2.424 | 3.599 |
| T2M Guo et al. (2022a) | 0.457 | 0.639 | 0.740 | 1.067 | 3.340 | 9.188 | 2.090 | 2.770 |
| MDM Tevet et al. (2022) | - | - | 0.611 | 0.544 | 5.566 | 9.559 | **2.799** | 0.497 |
| † MDM Tevet et al. (2022) | 0.405 | 0.591 | 0.701 | 0.825 | 3.619 | 9.411 | 2.497 | 1.147 |
| MotionDiffuse Zhang et al. (2022) | 0.491 | 0.681 | 0.782 | 0.630 | 3.113 | 9.410 | 1.553 | 1.954 |
| T2M-GPT Zhang et al. (2023a) | 0.491 | 0.680 | 0.775 | 0.116 | 3.118 | 9.761 | 1.856 | 0.514 |
| MLD Chen et al. (2022) | 0.481 | 0.673 | 0.772 | 0.473 | 3.196 | 9.742 | 2.413 | 0.404 |
| MotionGPT Jiang et al. (2023) | 0.492 | 0.681 | 0.778 | 0.232 | 3.096 | **9.528** | 2.008 | - |
| Ours | **0.516** | **0.709** | **0.805** | **0.070** | **2.948** | 9.655 | 1.835 | **0.376** |

layers $m = 8$, the dimension of transformer input is 512, the dimension of transformer's hidden layer is 1024, the number of transformer heads is 4 and the guidance-scale parameter for classifier-free task $s = 2.5$. It is noted that we adopt the Exponential Moving Average(EMA) training trick to help get more stable results. Besides, we train 300K iterations with batch size 512 on HumanML3D and 100K with batch size 256 on KIT-ML. By the way, our training stage costs about 43 hours on 4 A100 GPUs on HumanML3D.

## 4.2 COMPARISONS WITH STATE-OF-THE-ARTS

**Experimental Results on HumanML3D** The quantitative results of our experiments on Hu-manML3D are listed in Tab.1. According to the table, our method achieves the best results against previous methods on almost all evaluation metrics, which implies that synthesized motion sequences by our method are closest to the ground-truth description and ground-truth motion sequences. It is noted that the metrics rely on the pre-trained embedding model, so it is normal that our results of R-Precision are a little higher than the reported results of ground truth.

**Experimental Results on KIT-ML** Our experimental results on the KIT-ML dataset are listed in Tab.1 last column. The evaluation settings are the same as experiments on HumanML3D. The number of joints of the KIT-ML dataset is different from that of HumanML3D so we do not conduct semantic enhancement on KIT-ML. Therefore, we only report the results of FID with our basic CAMD model. According to the table, we also achieve the best results.

Table 2: Ablation studies and semantic enhancement evaluation of our method on HumanML3D. We short Dynamic-Enrich Features Encoder as DEFE, and Semantically Aligned Decoder as SAD. We evaluate the performance of our semantic enhancement with additional metrics translation similarity(TS), head-orientation similarity(HOS), and left forearm similarity(LFS). We bold the best results and underline the second ones.

| Methods | R-Precision ↑ | | | FID ↓ | MM-Dist ↓ | TS ↑ | HOS ↑ | LFS ↑ |
|---|---|---|---|---|---|---|---|---|
| | Top-1 | Top-2 | Top-3 | | | | | |
| Real | 0.511 | 0.703 | 0.797 | 0.002 | 2.974 | 1.000 | 1.000 | 1.000 |
| MDM Tevet et al. (2022) | 0.405 | 0.591 | 0.701 | 0.825 | 3.619 | 0.657 | 0.541 | 0.385 |
| Ours (DEFE only) | 0.494 | 0.688 | 0.788 | 0.342 | 3.108 | 0.691 | 0.625 | 0.489 |
| Ours (SAD only) | **0.530** | **0.719** | **0.812** | 0.105 | **2.903** | 0.695 | 0.625 | 0.532 |
| Ours (CAMD only) | 0.510 | 0.700 | 0.795 | 0.098 | 2.971 | 0.689 | 0.654 | 0.557 |
| Ours (full) | 0.516 | 0.706 | 0.805 | **0.070** | 2.948 | **0.725** | **0.683** | **0.620** |

**Ablation Studies** The experimental results of ablation studies have been listed in Tab.2. According to the table, with a single Semantically Aligned Decoder module, our method has achieved great performance. However, the module may be easy to overfit to obvious features of motion such as

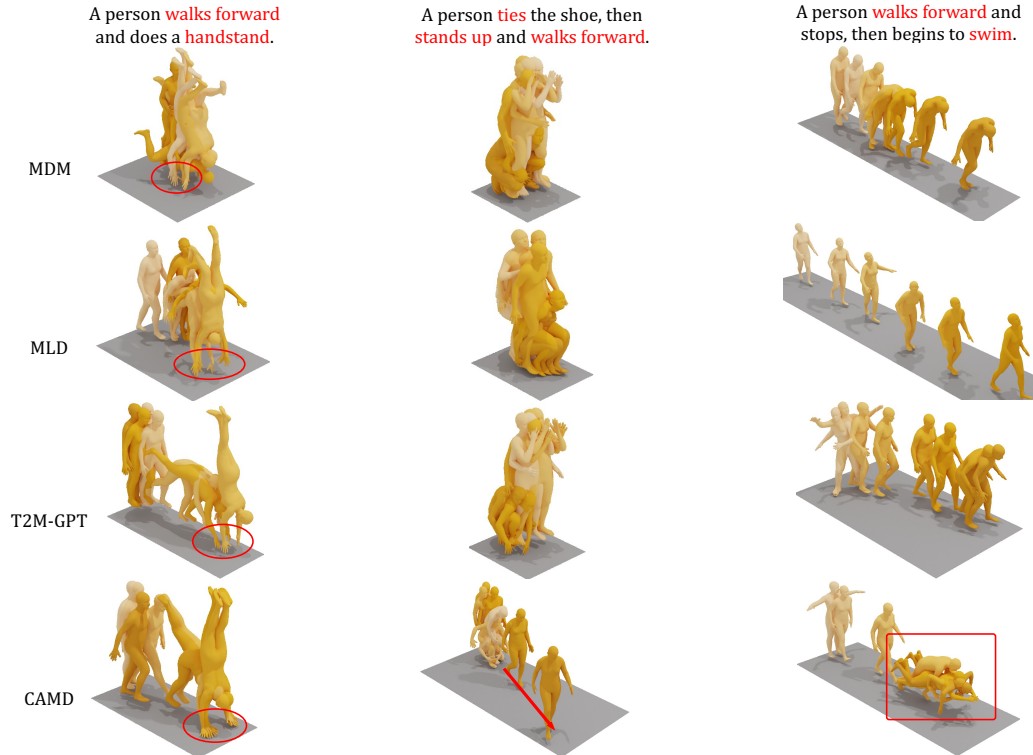

Figure 4: Qualitative comparison of the state-of-the-art approaches. We visualize motion sequences synthesized with three textual descriptions. We mark the main difference with red color. The results of our method align better with the descriptions than other methods.

translation, and ignore the specific body parts such as the head and left forearm. As a supplement, Dynamic-Enrich Features Encoder captures better global motion information. With both optimizations, the synthesized data distribution is closer to ground truth according to the table. Then, we try to synthesize motion sequences in the test set of HumanML3D with our enhanced textual descriptions. The results imply that our model synthesizes closer results to ground truth motion data with enhanced textual descriptions than original textual annotations.

**Semantic Enhancement Evaluation**   To observe the control effects after semantic enhancement on different body parts more intuitively, we evaluate the additional metrics in Tab.2. According to the table, though the synthesized results get better as the model optimizations, there is still an obvious gap. To address the problem, our semantic enhancement efficiently improves the control ability for enhanced parts and meanwhile improves the whole-body similarity FID between generated motion and ground truth ones.

## 4.3   VISUALIZATION

In this section, we visualize several representative demos synthesized with general descriptions by different methods. Then, we synthesize demos with enhanced descriptions of translation, left forearm positions, and head orientation separately.

**General Descriptions Synthesis**   We synthesize a few representative descriptions with four models: MDM Tevet et al. (2022), MLD Chen et al. (2022) T2M-GPT Zhang et al. (2023a), and our method SemanticBoost in Fig.4. The figure shows that motion sequences synthesized by our approach align the textual descriptions better than other methods. According to the results of the first column, with our refined representation and optimized structure, our method can more accurately synthesize the special joints with few corresponding descriptions in the dataset, such as the wrist.

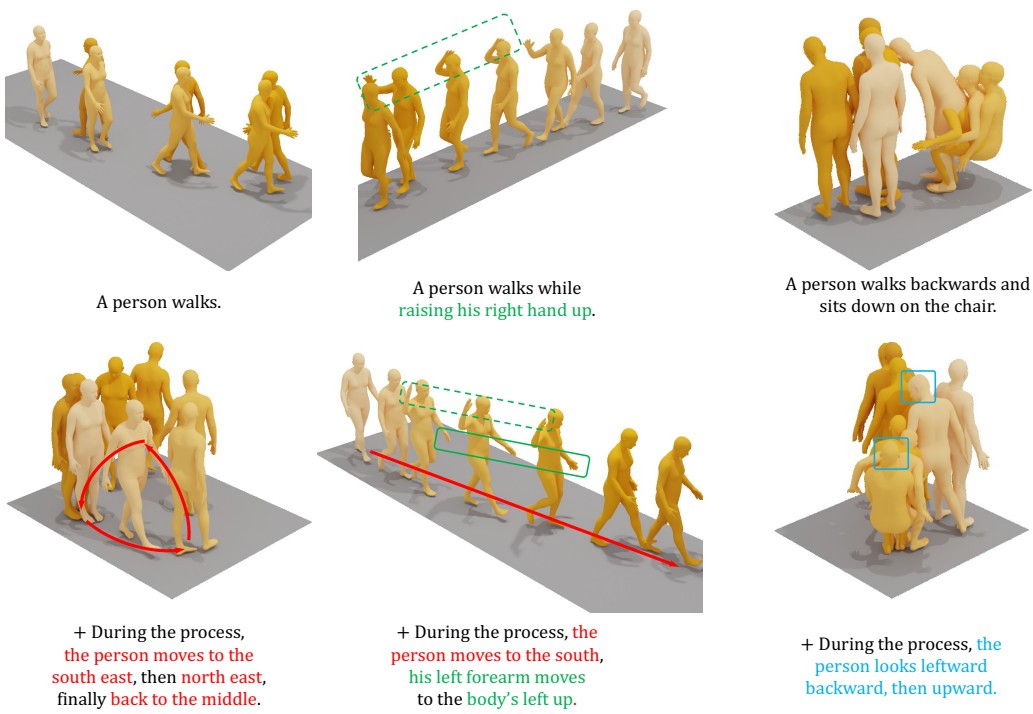

Figure 5: Synthesized motion sequences with and without enhanced descriptions. We emphasized additional translation descriptions in the first column, left forearm descriptions in the second column, and head orientation descriptions in the third column of the figure, respectively. We mark the corresponding results with red, green, and blue.

Additionally, the performance of synthesizing uncommon mixed actions such as "tie" and "walk", "walk" and "swim" also demonstrates the capability of our optimized model to synthesize complex textual descriptions.

**Enhanced Descriptions Synthesis** In Fig.5, we demonstrate the performance with and without semantic enhancement descriptions. The previous works may control the forward direction by rotating the motion sequences. To demonstrate the unique control capability of our method, we conducted continuous orientation control in the first column of the figure. The results show that our method enables continuous orientation control, which cannot be performed with simple rotation. Then, in the second column of the figure, we synthesize motion sequences with additional left forearm descriptions in the presence of interference from the right-hand action. The results align with the corresponding descriptions successfully. Finally, in the third column of the figure, we demonstrate the control capability of head orientation. Synthesized results demonstrate our performance again.

## 5 CONCLUSION

In this paper, we propose a framework Semantic Boost, to address the difficulties of generating motion from intricate and precise textual descriptions. Our framework consists of a Semantic Enhancement module and a Context-Attuned Motion Denoiser. The Semantic Enhancement module enhances the existing descriptions with extracted details from ground truth motion to generate descriptions that have better alignment with motion data. Our CAMD can understand longer and more complex semantic descriptions and synthesize motion that is aligned better with descriptions by capturing global motion information and utilizing word embeddings. The extensive experimental results demonstrate that our method achieves the best text-to-motion synthesis performance.

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
