# OpenReview forum: "SemanticBoost: Elevating Motion Generation with Augmented Textual Cues"
_ICLR.cc/2024/Conference — ICLR 2024 Conference Withdrawn Submission_

### Official Review · Reviewer_fDsx · 2023-10-14

**Soundness:** 2 fair
**Presentation:** 1 poor
**Contribution:** 2 fair
**Rating:** 5
**Confidence:** 5

**Summary:**

# I took about 12 hours to handle this submission. I list my confident reviews as follows.

Existing techniques often rely on large language models but struggle with precise alignment between text and motion data, leading to suboptimal results. This paper tries to enhance the motion generation quality from the aspect of text. The authors introduce a framework called SemanticBoost to support the motivation.

**Strengths:**

+ Good motivation to enhance the text quality.

+ Provide a demo video.

+ Organizing of the paragraphs is good.

**Weaknesses:**

- Poor writing. I list some cases as follows. **In the rebuttal session, I suggest authors revise all issues and list them on the openreview platform. I will check them one by one.**
    - In this submission, there are full of incorrect citation format. For example, in the first sentence of the introduction, Zhang et al. (2023a) should be (Zhang et al. 2023a). The authors seem not clear about the difference between `\cite` and `\citep`.
    - Orphan row. It is unprofessional, which seems trying to make up 9 pages. Like the final line of the introduction.
    - Colloquial expressions are unprofessional. Case: “**By the way**, our training stage … …”.
    - SemanticBoost or Semantic Boost? This is an inconsistent statement.
    - “Motiongpt: Human motion as a foreign language” has been accepted by NeurIPS 2023. Please revise the citation. **Please list all similar issues and revise them.**
    - Incorrect use of quotation marks. For example, in page 4, “forward” should be ``forward''.
    - Incorrect use of comma marks. The comma before "which" is wrong.
    - “… the diffusion-based generative method, such as MDM Tevet et al. (2022), MotionDiffuse Zhang et al. (2022) and faster MLD Chen et al. (2022) …”. What does “faster” mean? Faster in inference or training? It is confusing.
    - Missing explanation of equations. In Eq. 1, what do $T_i$ and $p^{mid-ear}$ mean? This makes readers confused. Does $Z$ mean $Z+$?
- Missing discussion on some motion diffusion models. I list some related work.

    [1]: Xu, Sirui, et al. "InterDiff: Generating 3D Human-Object Interactions with Physics-Informed Diffusion." *ICCV,* 2023.

    [2]: Yuan, Ye, et al. "Physdiff: Physics-guided human motion diffusion model." *ICCV, 2023*.

    [3]: Chen, Ling-Hao, et al. "HumanMAC: Masked Motion Completion for Human Motion Prediction." ICCV, 2023.

    [4]: Barquero, German, Sergio Escalera, and Cristina Palmero. "Belfusion: Latent diffusion for behavior-driven human motion prediction." *ICCV,* 2023.

    [5]: Shafir, Yonatan, et al. "Human motion diffusion as a generative prior." *arXiv preprint arXiv:2303.01418* (2023).

- **Unfair comparison. (I flag this submission as an ethical issue on fairness concerns and should be reviewed by an Ethics Reviewer.)**
    - As discussed in the “motion representations” part, the authors take the 269-dim motion representation, which is different from baselines. Authors should report all baseline results with 269-dim motion representation.
    - Why is the 269-dim motion feature better than the 263-dim feature? Please provide experiment ablation to verify it.
    - Authors do not provide mean and standard values of results. Baselines report the mean and standard values with 3 times repeating.
- In section 2.2, the authors claim that “However, the improved text annotations may not effectively align with the ground-truth motion data”. Any experimental results to support it?
- In ablation, authors introduce TS, HOS, and LFS metrics. Any explanation? Any citation? Why do not compare it with the baselines on Tab. 1?
- **Missing results on KIT-ML.**
    - Why does the joint number affect the experiments? If so, the method proposed in the paper will have significant limitations. Because it only applies to specific skeleton structures.
    - If it affects the experiment, why not use motion retargeting methods?
- Missing discussion on limitation.
- **Missing ablation.**
    - In Tab. 2, authors show the ablations on three components respectively. No results on the following settings: none of three, and with two components. The combination of ablation experiments should be 8 groups. Any missing ablation should explain why.
    - No visualization result to support the ablation in Tab. 2. This makes it unclear how exactly any of these components benefit the generated results.
- Authors do not provide any codes and appendix to support the reproduction. This makes it hard to follow. Note that this is the reason to reject a paper, but it reduces the value of a work to the community.
- **Marginal contribution.** The semantic enhancement method only supports providing the direction of motion, which is only a small part of the motion. Besides, motions also include action counting, body-part motion, and something else. If the augmented text cues can resolve these, I suggest authors provide the following cases.
    - “A person jumps three times.”
    - “A person jumps four times.”
    - “A person lifts his left leg then walks.”
    - “A person lifts his right leg then walks.”
- The figure of DEFE is confusing. Please detail the network architecture.
- For network architecture, the names are complex, which makes it hard to read. Additionally, authors sometimes use their full names and sometimes use their abbreviated names. The statement here is hard to follow.

**Questions:**

Question:

- How many words are in the status table? Authors should list the implementation details as detailed as possible.
- What will the results be if there is no text enhancement?
- **I am very confused about why the R-Precision is higher than GT. This is my main concern. Do authors use the augmented text for the test? If so, can authors report the R-Precision with original texts?**
- Does the feature extractor for calculating R-Precision use the checkpoint provided by the HumanML3D paper or the model is trained by the authors?

---

## Rating

My concerns come from the unfair comparison, poor writing, missing experiments, contribution, and metrics. I hope these concerns can be resolved. Now, I think this submission is below the borderline. After the rebuttal and discussion, I will revise my rating to a clear rating (reject or accept).

**Details Of Ethics Concerns:**

See the weakness part on **Unfair comparison**

---

### Official Review · Reviewer_XYJJ · 2023-10-20

**Soundness:** 2 fair
**Presentation:** 2 fair
**Contribution:** 2 fair
**Rating:** 3
**Confidence:** 4

**Summary:**

This paper proposed a semantic enhancement module and a context-attuned motion denoiser to address the problem of insufficient semantic annotation and weak context understanding for text-driven human motion generation. Extensive experiments show the effectiveness of the proposed method.

**Strengths:**

1. The quantitative evaluation of control capabilities in this paper is interesting.
2. This work is easy to follow and most of the techniques are correct.
3. The models show comparable even better results than previous methods.

**Weaknesses:**

1. Technical Contributions: In fact, the conceptions of semantic enhancement and context understanding structures are not new in the current generative models field. The authors mainly introduce a simple semantic annotation augmentation method. But this theory has already been proposed in Action-GPT [1]. Thus, the technical contributions are very limited in this work, considering most of the used modules are from existing works. If there are any insightful contributions, the authors can highlight them and compare them with previous works.

2. More Motivations: In this paper, east, south, west, north, and other directions are used to define the direction of the body. But there are only left and right directions in the motion space, and there is no definition of east, south, west, north, etc., therefore additional explanations are required.

**Questions:**

1. Although enhancing the text in the training set can enhance the correspondence between the motion description and the motion, it also increases the length of the text. However, the text in the test set is not enhanced, which means that the distribution of motion descriptions between the training set and the test set may differ. How did you solve the problem?

2. Due to the randomness of the generated models, most works run multiple times and report a 95% confidence interval. It is better to provide results for multiple runs as well.

---

### Official Review · Reviewer_TxQT · 2023-10-29

**Soundness:** 2 fair
**Presentation:** 2 fair
**Contribution:** 2 fair
**Rating:** 5
**Confidence:** 4

**Summary:**

The authors automatically extract semantics from raw motion sequences, thus boosting the text annotation for text-to-motion tasks.
To exploit the extra semantic information, a Context-Attuned Motion Denoiser is proposed.
By adding a global motion dynamic feature, CAMD is capable of utilizing the global information of the sequence.
CAMD also utilizes word embeddings to enhance the alignment between input motion and input text prompts.
Impressive generation quality is achieved by the newly proposed method.

**Strengths:**

- The idea of digging out inherent semantic information from motion data itself with no human annotation is interesting and worth exploration.

- Incorporating a global feature for the denoising process is a useful technical trick.

- The newly proposed metrics are interesting supplements, which could be viewed as a decomposed and individual version of FID, evaluating the distribution similarity between individual motions.

- The generation quality is impressive in the demo video.

**Weaknesses:**

- A highly related citation PoseScript[1] is missed. The author should clarify the difference between PoseScript and the proposed Semantic Enhancement Module.

- Some definitions in the Semantic Enhancement Module are not clarified well. It would be better to provide formulations for the enhancement of hand-hip relationships and body directions as in Eq. 1.  Also, I did not get why rotating the motion sequences around the vertical as augmentation is useful.

- The idea of incorporating a global token for motion generation is similar to NeMF[2], which is a missed citation.

- The higher R-Precision than GT is a little unsettling. Since an imperfect pre-trained embedding model is adopted, there is a small possibility of over-fitting. The result of the SAD-only model, with a more significant advantage over GT on R-Precision and MM-Dist, seems to advocate the hypothesis of over-fitting since SAD is specifically devised to boost motion-text alignment. In view of this, the automatic metrics seem to be not enough for evaluation. If applicable, extensive user studies would be preferable.

- The ablation studies are only conducted for different model structures, while the key contribution, the Semantic Enhancement (Sec. 3.1), is not well discussed. It seems that with Semantic Enhancement and the DEFE-only model, only a minor improvement is achieved compared to the similar diffusion-based method MotionDiffuse. Though the visualizations provide impressive comparisons, it would be better with quantitative results.

- It is claimed that the newly proposed metric reflects the control ability of models, but I did not see this. These metrics are still based on comparison with GT samples, instead of the input control signals (text prompts). Therefore, I understand them as a decomposed and individual version of FID, as they focus on the distribution of body part movements, and are computed at sequence-level instead of dataset-level.

[1] Delmas G, Weinzaepfel P, Lucas T, et al. PoseScript: 3D human poses from natural language[C]//European Conference on Computer Vision. Cham: Springer Nature Switzerland, 2022: 346-362.

[2] He C, Saito J, Zachary J, et al. Nemf: Neural motion fields for kinematic animation[J]. Advances in Neural Information Processing Systems, 2022, 35: 4244-4256.

**Questions:**

- For Semantic Enhancement in Sec. 3.1, I did not understand the body direction status ``east, north, west, south''. How are these statuses defined? As I understand, these four cardinal directions seem less meaningful in the canonical motion space. Would they suffer from the potential misalignment between the definition in this project and in the real world?

- Would the down-sample rate of 10 be too high in Sec. 3.1, especially for the hand-hip relationship? Some status changes might be omitted.

- How are the enhanced text descriptions aggregated for training? How many descriptions are auto-generated per sequence? It appears that the auto-generated descriptions are much longer than the original descriptions. Are they simply truncated at the length of 77?

---

### Official Review · Reviewer_qEjh · 2023-11-01

**Soundness:** 2 fair
**Presentation:** 2 fair
**Contribution:** 2 fair
**Rating:** 3
**Confidence:** 5

**Summary:**

This paper introduces a new framework for text-driven motion generation, with a primary focus on enhancing feature extraction from textual descriptions. It achieves this through two key steps. First, it designs a pipeline to automatically enhance textual descriptions by adding more detail. Second, it introduces new submodules to effectively extract information from these enhanced textual details.

**Strengths:**

1. The proposed method establishes a new state-of-the-art.

2. The paper is well-written, ensuring that its content is easily understandable for readers.

**Weaknesses:**

My primary concerns regarding this paper are related to the limited extent of experimental comparisons and analyses.

1. The paper lacks a comparison with ReMoDiffuse\[1\], which demonstrates significantly higher accuracy on the KIT-ML dataset compared to this paper's results. Additionally, the claim in the paper stating, 'Notably, our method is the first diffusion-based approach to outperform autoregressive model-based approaches, achieving state-of-the-art performance on the challenging HumanML3D dataset,' is inaccurate. ReMoDiffuse also outperforms T2M-GPT, so this paper is not the first to do so.

2. While the paper provides an ablation study of the overall modules, many similar designs are found in the literature. If the authors want to prove the effectiveness of their proposed modules, they should compare these modules with other designs. For example, MotionDiffuse uses a complete text feature sequence as the input for cross-attention, and ReMoDiffuse employs a similar design. The authors need to compare their Context-Attuned Motion Denoiser with these two to demonstrate the applicability of their design.

3. The paper builds upon MDM with various additional modules. The authors should provide information about the parameter count and inference time for their method, especially when compared to MDM. If their method has more parameters compared to MDM, the authors should consider increasing the number of layers or latent dimensions in MDM for a fair comparison to validate that the improvement in results is not solely due to an increase in the number of computational cost.

4. The paper's experiments related to text enhancement are limited, with only an overall accuracy comparison. The authors should provide more analysis, such as which steps are crucial during the enhancement process, and how they impact accuracy or visual results.

5. The authors should provide user studys to quantitatively measure the generation quality.

\[1\] Zhang et al. ReMoDiffuse: Retrieval-Augmented Motion Diffusion Model

**Questions:**

Please kindly refer to the weaknesses mentioned above.